# The Role of L-Arginine-NO System in Female Reproduction: A Narrative Review

**DOI:** 10.3390/ijms232314908

**Published:** 2022-11-28

**Authors:** Jozsef Bodis, Balint Farkas, Bernadett Nagy, Kalman Kovacs, Endre Sulyok

**Affiliations:** 1Department of Obstetrics and Gynecology, University of Pecs School of Medicine, 7624 Pécs, Hungary; 2MTA-PTE Human Reproduction Scientific Research Group, University of Pécs, 7624 Pécs, Hungary; 3National Laboratory on Human Reproduction, University of Pécs, 7622 Pécs, Hungary; 4Faculty of Health Sciences, University of Pécs, 7621 Pécs, Hungary

**Keywords:** l-arginine, methylarginines, nitric oxide, oxidative stress, female reproduction

## Abstract

Accumulating evidence are available on the involvement of l-arginine-nitric oxide (NO) system in complex biological processes and numerous clinical conditions. Particular attention was made to reveal the association of l-arginine and methylarginines to outcome measures of women undergoing in vitro fertilization (IVF). This review attempts to summarize the expression and function of the essential elements of this system with particular reference to the different stages of female reproduction. A literature search was performed on the PubMed and Google Scholar systems. Publications were selected for evaluation according to the results presented in the Abstract. The regulatory role of NO during the period of folliculogenesis, oocyte maturation, fertilization, embryogenesis, implantation, placentation, pregnancy, and delivery was surveyed. The major aspects of cellular l-arginine uptake via cationic amino acid transporters (CATs), arginine catabolism by nitric oxide synthases (NOSs) to NO and l-citrulline and by arginase to ornithine, and polyamines are presented. The importance of NOS inhibition by methylated arginines and the redox-sensitive elements of the process of NO generation are also shown. The l-arginine-NO system plays a crucial role in all stages of female reproduction. Insufficiently low or excessively high rates of NO generation may have adverse influences on IVF outcome.

## 1. Introduction

Since the discovery of the l-arginine-NO system, great progress has been made in our understanding of its physiological role and clinical implications [1]. Most studies have been published in the field of cardiology, nephrology, neurology, neuroendocrinology, immunology/inflammation, and fertility in an attempt to delineate NO involvement in complex, interacting biological processes. In addition to its major role in maintaining vascular homeostasis by controlling vascular tone, by inhibiting platelet and leukocyte /monocyte adhesion to the endothelium, as well as by inhibiting the growth and proliferation of vascular smooth muscle cells, NO exerts cell/tissue-specific autocrine/paracrine function [2]. Moreover, NO/NOS proved to be a critical player in signal transduction and transcription factor regulation.

In the present review, we made an attempt to summarize the major aspects of the function of the l-arginine-NO system, from cellular l-arginine uptake to l-arginine catabolism by NOS to NO, arginase to l-ornithine and downstream metabolites, arginine methylation, and l-arginine-NO related oxidative stress. Particular attention was given to the relevance of the l-arginine-NO system in female reproduction and the debate on l-arginine supplementation to improve IVF success, and pregnancy outcomes were briefly discussed.

## 2. Clinical and Physiological Significance of L-Arginine Methabolism

### 2.1. L-Arginine Transport into the Cells

Amino acid uptake is an essential part of cellular metabolism and required for protein synthesis and numerous enzymatic reactions. Associations have been proven between the turnover rate of amino acids and the developmental potential of oocytes and early embryos. Cellular uptake of individual amino acids or a certain group of amino acids is achieved by separate, well-defined transporters that are encoded by different genes. L-arginine is transported by members of the cationic amino acid transporter (CAT) family. This transport system is Na^+^-independent, pH-insensitive, and activated by hyper-polarization and substrate trans-stimulation. CAT-1, CAT-2B, and CAT-3 isoforms are involved in arginine transport that are encoded by the respective genes of the solute carrier (SLC) gene families. CAT-2A is a split variant without transport activity. Isoform 1 has been identified as a low-affinity, high-capacity transporter that enhances the cellular uptake of cationic arginine, lysine, and ornithine, while isoform 2B mediates the transport of these same amino acids but with much higher affinity for arginine than isoform 1 [3].

The major characteristics of amino acid transport systems in early mouse embryo development have been explored by Van Winkle et al. It has been demonstrated that these systems are developmentally regulated to meet the requirements of rapid growth and differentiation. Accordingly, the most highly expressed transport proteins for arginine have been identified in oocytes (CAT-2, CAT-1), in two-or-eight-cell embryo and blastocyste (CAT-1/CAT-2). It was supposed that the co-expression of CAT-1 and CAT-2 might be accounted for by at least two distinct transport activities of the superfamily they belong to. The abundance of corresponding mRNAs expression has also been demonstrated in mouse embryos at different stages of preimplantation development. As the same amino acid transport systems are operating in humans, it is reasonable to assume that the developmental pattern seen in mouse embryos may apply for human preimplantation embryos [4]. In support of this notion, microarray studies of human oocytes revealed the presence of genes encoding members of the transport systems mediating the cellular uptake of l-arginine [5]. Furthermore, mRNAs for cationic amino acid transporters (SLC7A1, SLC7A2, and SLC7A3) have been detected in ovine uterus epithelia and in the trophectoderm and endoderm of peri-implantation conceptus. The abundance of SLC7A1 and SLC7A2 has been enhanced by the estrous cycle and pregnancy and progesterone treatment, whereas that of SLC7A3 proved to be unaffected by either of these conditions [6].

Importantly, arginine transport is also accomplished via the B+o transport systems that mediate cellular accumulation of leucine, tryptophan, and arginine, with particular preference to arginine. These systems are operating already in the early embryonic development, therefore, they have been claimed to deplete amino acids from uterine secretions, to suppress T cell proliferation, and to protect implanted embryos from rejection [4].

With respect to tryptophan uptake by B+ transport system, it is to be stressed that this essential amino acid is the substrate of two competing metabolic pathways: the tryptophan-kynurenine and the tryptophan-serotonin (5-HT) pathways [7]. Convincing evidence have been provided for the essential role of tryptophan catabolism to both 5-HT and kynurenine in oocyte maturation, fertilization, implantation, and early embryonic/fetal development. In fact, embryo viability has been shown to be enhanced through 5-HT signaling as the paracrine/autocrine function of the serotoninergic network is required in the earliest embryonic development [7,8,9,10]. On the other hand, activation of kynurenine pathways resulted in decreased number of CD45-positive leukocytes and provided a possible immunological mechanism to establish embryo tolerance in early pregnancy [11].

The immune protection of embryo by kynurenines is further supported by the observations that with progressing pregnancy, IDO (indoleamine-2,3-dioxygenase, the enzyme initiating tryptophan catabolism) expression is up-regulated by inflammatory cytokines including its most potent stimulant, interferon gamma, and immunosuppressive kynurenines are generated [12]. Conversely, IDO inhibition with methyl-tryptophan, or deletion of IDO gene caused pregnancy complications and fetal compromise [13,14]. It can be concluded, therefore, that adequate tryptophan supply for the generation of 5-HT and kynurenines is critical for the success of reproduction, and the two metabolic pathways should be kept in balance, although in IVF patients the tryptophan-5-HT pathways prevailed over tryptophan-kynurenine pathway when chemical/clinical pregnancy could be achieved [7].

### 2.2. The Process of Arginine Methylation

NO generation is mainly controlled by methylarginines. Asymmetric dimethylarginine (ADMA) and monomethylarginine (MMA) competitively inhibit NOS isoforms, while MMA and symmetric dimethylarginine (SDMA) inhibit cellular uptake of l-arginine by cationic amino acid transporters [15] (Figure 1). Methyl groups for the methylation of arginine residues of proteins are provided by the folate-dependent homocysteine/methionine cycle. In this metabolic pathway, methionine is initially activated by ATP to S-adenosylmethionine (SAM) that serves as methyl donor for methyltransferases to add methyl groups to substrates including proteins, histones, DNAs and RNAs [16].

After methyl transfer, SAM is converted to S-adenosylhomocysteine (SAH) that may undergo hydrolysis to homocysteine and adenosine. To prevent homocysteine accumulation, it is remethylated to methionine by the vitamin B12- and folate-dependent enzymes; methylene tetrahydrofolate reductase and methionine synthase (Figure 1). Inadequate dietary folate supply and/or gene polymorphisms of the remethylation enzymes may compromise the function of methionine/homocysteine circle with subsequent elevation of plasma homocysteine levels, and reduced methyl group generation for transmethylation reaction that may be associated with adverse clinical consequences [17,18,19].

With respect to female reproduction, folate insufficiency and homocysteine excess were found to result in multiple developmental abnormalities, recurrent early miscarriage, pre-eclampsia, and low birth weight. Folate/vitamin B12 supplement proved to reduce these complications by overcoming methionine trap and by re-establishing the functional integrity of the methionine /homocysteine circle [16,17,18,19,20].

Concerning NO production, inhibitory methylarginines are formed by protein arginine methyltransferases (type I PRMT and type II PRMT) via immediate precursor protein MMA, then it is catabolized to ADMA (type I PRMT) and SDMA (type II PRMT). After proteolysis, these methylarginines are released, and in their free form they exert their biological action. Accelerated proteolysis of proteins with methylated arginine residues and/or their reduced elimination may result in the accumulation of methylarginines. ADMA and MMA are mainly metabolized by dimethylarginine dimethylaminohydrolases (DDAHs) to dimethylamine and citrulline, while SDMA is removed by urinary excretion [15].

A great body of evidence indicates that methylarginines, especially ADMA and the integrated index of arginine methylation (arg-MI), is a well-established marker and mediator of the progression of cardiovascular and renal diseases [21]. With these observations in line, our group reported significant inverse relationships of the number of oocytes retrieved and that of the embryos conceived to follicular fluid l-arginine, ADMA, SDMA, and MMA levels, respectively, in women undergoing IVF. Furthermore, higher FF methylarginine levels had negative impact on IVF outcome in terms of pregnancy rate. Importantly, no differences were noted in the l-arginine/ADMA ratio, an estimate of NO bioavailability, between groups of different IVF outcome [22].

These seemingly conflicting findings can be reconciled by assuming that to a certain extent, methylated arginines may influence the reproductive performance independent of NOS inhibition and NO generation. Consistent with this notion, methylarginines have also been shown to act beyond direct NOS inhibition in patients undergoing cardiac evaluation [21]. The concern about the high rate of methylation is further emphasized by the observations that members of PRMT family mediate the methylation of arginine residues in several nuclear and cytoplasmic protein substrates. Therefore, it may interfere with multiple cellular functions including signal transduction, transcription factor activation, RNA splicing, chromatin remodeling, DNA damage repairs, and protein-protein interaction [23,24].

The involvement of PRMTs in oocyte maturation and early embryo development has been well-documented. Altered patterns of their ovarian gene expression and dysregulation of PRMTs mediating post-translational modifications of histone protein and DNA may be associated with poor quality oocytes and compromised developmental potential. In support of this notion, the essential role of PRMTs in genome maintenance, cell proliferation, folliculogenesis, and post-implantation development has been demonstrated [25,26,27,28,29].

A recent mRNA-seq and genome-wide DNA methylation study of human ovarian granulosa cells demonstrated significant non-random changes in transcriptome and DNA methylome features as women age and their ovarian functions deteriorate. Increased methylation in highly methylated regions and decreased methylation in poorly methylated regions were equally associated with age-related decline in ovarian function [30]. Methylation of arginine residues of histone and non-histone proteins are also thought to be an important regulator of cellular functions, in particular, the structure and function of DNA, so it may also contribute to epigenetic modifications [30] (Figure 2).

Taken together, methyl groups are generated as a normal product of cellular metabolism and have a regulatory role in several cellular processes. However, both a low rate and the excessively high rate of methylation have the potential to cause cellular dysfunction, to interfere with the finely tuned, complex interactions of metabolic pathways related to or independent of NO production. As a result, it may compromise healthy development of oocytes and embryos with subsequent pregnancy failure/complications. Unfortunately, the narrow range of methylation optimum has not been defined, therefore, efforts should be made to determine the upper and lower limit of methylation normality outside of which higher risk of fertilization and pregnancy success can be anticipated.

### 2.3. L-Arginine-Arginase Pathway

Arginase is a urea cycle enzyme that hydrolyses l-arginine to urea and l-ornithine. Two distinct arginase isoforms, arginase I and arginase II, have been identified with different tissue distribution, cellular location, and immunoreactivity. The two isoforms are encoded by different genes. The cytosolic arginase type I is mainly expressed in the liver and involved in urea synthesis, while mitochondrial type II is widely distributed and plays a critical role in regulating NO synthesis [31,32,33].

Importantly, arginase competes with NOS for their common substrate, l-arginine, so it may reduce l-arginine bioavailability by redirecting l-arginine catabolism from NO to l-ornithine, which is the precursor of polyamines and proline (Figure 2). It has been well-established that these substrates are intimately involved in various levels of reproductive processes, from folliculogenesis to clinical pregnancies [34]. Furthermore, arginase causes NOS uncoupling, resulting in superoxide and peroxynitrite generation, which further compromises NOS activity [35,36]. It is also to be considered that arginase is co-expressed with NOS in endothelial cells suggesting the mutual tight control of their activity [37].

Concerning the involvement of arginase in regulating reproductive functions, it has been demonstrated in reproductive organs, in particular in the ovarian structures, that it plays a role in the production of ornithine which can be metabolized to polyamines that are required for cell division, proliferation, and differentiation [38,39].

A comprehensive review by Lefevre et al. provides a detailed outline of the critical role of polyamines (putrescine, spermidine, spermide) in female reproduction. Experimental evidence are given indicating that they are needed for oogenesis, embryogenesis, implantation, and placentation. The hormonal regulation of polyamine synthesis and the association of polyamines with ovarian steroid hormones are also presented. These results were obtained by genetic and pharmacological manipulations of ornithine decarboxylase (ODCI), the rate-limiting enzyme of polyamine synthesis, and the antizyme (AZI) family members modulating the activity of ODCI [34] (Figure 3).

### 2.4. Role of NO in Female Reproduction

The endothelium-derived relaxing factor (NO) is a cellular messenger and effector molecule that participates in the control of a series of female reproductive processes. This gaseous molecule is highly reactive, unstable with very short half-life, and diffuses readily within the cells and into the neighboring cells. It is generated by NO synthases (NOSs). Three NOS isoforms have been identified: the constitutional neuronal (nNOS) and endothelial (eNOS) that are calcium/calmodulin-dependent, and the inducible (iNOS) that is calcium independent and induced by inflammatory cytokines/immune mediators. NOSs catalyze the reaction: l-arginine + O_2_ → NO + citrulline. The resulting NO activates soluble guanylate cyclase (sGC), increases cGMP concentration, and induces the activity of cGMP-dependent protein kinases, cGMP-gated ion channels and cGMP- regulated phosphodiesterase. Importantly, NO has been shown to have dual effects on reproductive processes depending on its cellular levels. NO exerts its physiological functions within a narrow range of cell/tissue NO levels. Both, excessive, or insufficient NO production may result in functional impairment and/or cellular damage. NOS isoforms are expressed in species-, cell- and maturational stage-specific manner [40,41].

The expression pattern of individual NOS has been established in various animal models and in humans during the whole period of the reproductive cycle including folliculogenesis, oocyte maturation, embryonic development, implantation, and pregnancy maintenance. The major observations of these studies have been recently reviewed in detail. Shortly, the major ovarian NOS isoforms are the eNOS and iNOS. Both have been detected in human granulosa and luteal cells and have been claimed to be involved in follicular development as stimulation with hCG increased their expression. Based on the cellular level of NO it may have different effects on developing follicles. High NO concentration proved to have anti-apoptotic effects and to protect cell survival, whereas decreased iNOS expression and low NO concentration induced apoptosis by activating the caspase-mediated cascade in in vitro cultured granulosa cells. Therefore, the alterations in NO generation of granulosa cells may result either in developing or in atretic follicles [40,41].

The involvement of l-arginine-NO system in the central regulation of reproduction has been documented by demonstrating the presence of neuronal NOS in the somatostatin neurons in the ventral medial nucleus of the hypothalamus [42]. In response to sustained elevation of estradiol levels eNOS containing neurons are activated and GnRH and LH surge occurs. On the other hand, inhibition of nNOS with intracerebroventricular l-NAME the estrogen induced LH surge was completely blocked implying the critical role of NO in the feedforward regulation of estradiol-LH axis [43].

NO has also been reported to have a regulatory role in steroidogenesis in granulosa and luteal cells of various species including humans. Namely, NO inhibited basal and gonadotropin-stimulated estradiol and progesterone secretion by inhibiting p450 aromatase activity and by down-regulation of its mRNA transcription [44,45]. In addition to the autocrine regulation of ovarian steroidogenesis by NO it has also been suggested to stimulate the secretion of hypothalamic gonadotropin-releasing hormone (GnRH) and the release of gonadotropins by pituitary cells [46,47].

The requirement of NO/NOS system for oocyte maturation, ovulation and luteinization, embryo development and trophoblast outgrowth has been established. However, evidence have been provided for the double role of NO in granulosa cells and oocytes depending on its concentration. High concentration of NO donor inhibited, while its low concentration stimulated meiotic resumption and resulted in higher rate of oocytes reaching metaphase II stage [48,49]. In this regard, it is also to be noted that while single-NOS knockout mice encountered no reproductive anomalies, double-knockout mice (iNOS/eNOS, eNOS/nNOS, iNOS/nNOS) were preferentially lost during early embryonic development [50,51,52,53]. Furthermore, administration of NOS inhibitors L-NA or L-NAME could induce developmental arrest of the embryo, but these inhibitory effects could be reversed by addition of an NO donor, or the second messenger cGMP analogues [50,54,55]. The essential role of NO/NOS in reproductive processes are further emphasized by the observations that the elevated levels of follicular fluid l-arginine and methylarginines were inversely related to the number of mature oocytes, viable embryos, and clinical pregnancies. These negative associations were assumed to be due to the reduction of NO generation by the NOS inhibitor endogenous methylarginines [24].

The NO/NOS system has been widely studied in normal pregnancies and in pregnant women presenting with preeclampsia. It has been confirmed that during pregnancy NO generation in the myometrium and placenta is markedly elevated and the high NO concentration has been implicated in maintaining uterus relaxation and low vascular resistance in utero-placental circulation. When the NO production is insufficient the foeto-placental circulation is compromised and pre-eclampsia with or without fetal growth retardation may develop. This pattern of NO generation may be causally related to the concomitant alterations of the endogenous NOS inhibitor ADMA, which was found to decease early in normal pregnancy and to increase in preeclampsia [56,57,58,59].

The pregnancy-related changes in ADMA [60] concentrations are thought to be accounted for by reduced expression of DDAH in trophoblast that control metabolic elimination of ADMA. It is also to be considered that beside the impaired function of l-arginine-NO system the up-regulation of l-arginine-arginase pathway may also be involved in the pathogenesis of pre-eclampsia [61].

The importance of NO in labor and delivery has also been documented. All three isoforms of NOS (eNOS, iNOS, nNOS) are expressed in the cervix and eNOS and iNOS expressed in the corpus of rat uterus. Furthermore, iNOS expression increased in the cervix, whereas it decreased in the corpus during labor indicating that its different regulation may be involved in the process of cervical ripening [62] (Figure 4).

### 2.5. Oxidative Stress and L-Arginine NO System

The role of oxidative stress in female reproduction has been extensively studied. Recent reports on the association between reactive oxygen species (ROS) and metabolic adaptation of oocytes and early embryos to the changes in their environment have shown ROS to affect oocyte developmental competence and subsequent embryo quality [63,64,65].

ROS are generated as a normal product of cellular metabolism and have a regulatory role in several cellular processes. When their excessive generation exceeds the capacity of antioxidant defense mechanisms oxidative stress ensues and ROS reacts with essential cellular elements causing cellular dysfunction, damage, and apoptosis. For oocyte maturation, successful fertilization and embryo formation physiological levels of ROS are needed, however, the optimum ROS levels have not been established. According to the “quiet metabolism” concept there are upper and lower limits of metabolic normality, outside of which embryo viability declines [66,67]. 

The present review attempts to summarize the impact of oxidative stress on various elements of l-arginine-NO system, and to present its relevance to female reproduction. Basic studies on this topic have been performed in vascular endothelial cells but the major messages appear to apply to the redox-sensitive reactions of NO generation and/or elimination during reproductive processes [68]. The oxygen-derived free radical superoxide rapidly reacts with NO and forms highly reactive intermediate, peroxynitrite which may cause oxidative damage to proteins, lipids, and DNA. In response to absolute or relative depletion of NOS substrate l-arginine NOS uncoupling occurs and the uncoupled NOS generates superoxide rather than NO [69]. Furthermore, endogenous (ADMA) or exogenous (L-NMMA) NOS inhibitor stimulates superoxide production by competing for the binding site of the enzyme thus limiting NO generation. The production of ADMA by PRMTs and its degradation by DDAH takes place in redox-sensitive fashion, therefore, the activation of these enzymes results in enhanced accumulation of the cellular ADMA pool [68]. ADMA and l-arginine analogues can further impair NO production by inhibiting cellular uptake of l-arginine through the cationic amino acid transporter and by the interaction of arginase with the l-arginine-NO system. This latter contention is supported by the observations that arginase over-expression depleted tissue l-arginine pool that redirected NOS to form superoxide anions which reduced NO bioavailability by generating peroxynitrite. Based on these findings the concept of feedforward regulation of arginase and peroxynitrite was developed implying that peroxynitrite up-regulates arginase which in turn generates more peroxynitrite that further compromise NO production [70]. ADMA and other methylarginines in follicular fluid have been claimed to be negatively associated with IVF success [24]. Others, however, failed to demonstrate any differences in plasma ADMA levels between the implantation positive and negative groups indicating that ADMA cannot be used as predictive marker of implantation success in IVF cycles [71].

### 2.6. L-Arginine Supplementation

In view of the relative or absolute l-arginine deficiency in clinical conditions with elevated ADMA levels it was relevant to assume that supplementation with exogenous l-arginine replenishes the tissue l-arginine store and restore the NOS/NO balance. In support of this concept the following mechanisms are to be considered; (a)l-arginine supplement may counteract the ADMA inhibition of the NOS-mediated NO production: (b) its cellular uptake via cationic amino acid transporter may be enhanced by mitigating the inhibitory effect of ADMA and (c) as an antioxidant it may reduce NOS-mediated superoxide production and may scavenge superoxide [72].

In spite of these theoretical considerations, the results of clinical trials with l-arginine supplementation are controversial, and its protective role has not been consistently proved. The reason for the inconsistencies of this approach is not apparent, although the possible role of “l-arginine paradox” has been proposed. This implies the dependence of cellular NO generation on exogenous l-arginine supply in spite of the calculated saturation of NOS with l-arginine [73].

In agreement with the potential of exogenous l-arginine to protect the functional integrity of l-arginine-NO system there have been reports on the l-arginine-related improvement of early embryonic development. Transcriptome analysis of porcine embryo culture medium revealed that its treatment with increasing concentrations of l-arginine increased the number of embryos developing to blastocyst stage and they have more total and trophoectoderm nuclei. High l-arginine concentration reduced the gene expression for cationic amino acid transporter (SLC7A1) but it left unaffected for protein arginine methyltransferases (PRMT1, PRMT3 and PRMT5). Furthermore, DDAH1 and DDAH2 message was differently regulated during development. In support of the essential role of the PRMT-DDAH-NO axis in development of preimplantation porcine embryos, DDAH1 null mutation proved to be lethal [74]. Supplementation the bovine embryo culture medium with l-arginine also favors preimplantation embryo development by improving embryo hatching rates and quality [75].

In humans, poor-responder patients undergoing IVF arginine treatment resulted in increased plasma and follicular fluid levels of arginine, citrulline and NO_2__/NO_3__ that was associated with increased number of oocytes retrieved and embryo transferred. Uterine and follicular Doppler flow improved indicating better ovarian response, endometrial receptivity, and the subsequent higher pregnancy rate. Moreover, l-arginine supplementation improved pregnancy outcome by reducing fetal loss, intrauterine growth restriction and pre-eclampsia [76,77] Additional evidence for increased endometrial receptivity by l-arginine was provided by the study showing that l-arginine added to the culture media at physiological or supra-physiological concentrations enhanced endometrial RL95-2 cell proliferation and reduced mitochondria-mediated apoptosis [78].

## 3. Conclusions

In spite of the beneficial effects of l-arginine supplement concerns are to be expressed about its routine clinical use. Namely, excess l-arginine may promote uncontrolled generation of NO and the related superoxide and peroxynitrite as well as the excessive formation of methylarginines that may further amplify the production of ROS. As a result, the finely tuned balance between NO generation and its complex control mechanisms may be disturbed with unwanted clinical consequences. Large-scale, randomized, double-blind, prospective studies are to be conducted to establish the safe timing, dose, and duration of l-arginine supplementation in female patients receiving care for reproductive disorders.

## Figures and Tables

**Figure 1 ijms-23-14908-f001:**
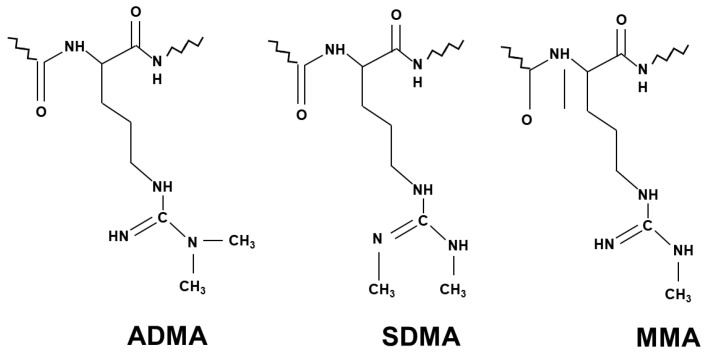
Chemical structure of the three different methylarginines.

**Figure 2 ijms-23-14908-f002:**
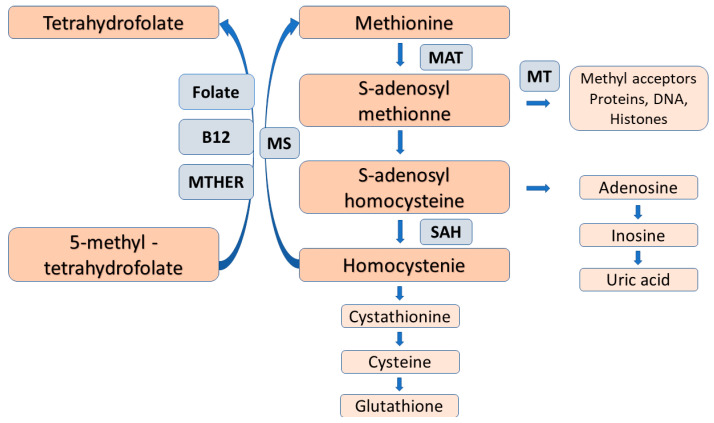
The folate-dependent methionine-homocysteine cycle. Trans-sulfuration of homocysteine for cysteine and glutathione, as well as folate deficiency reduce the reaction by methionine synthase and causes homocysteine accumulation with the subsequent inhibition of the conversion of 5-methyl-tetrahydroforate to the metabolically active folate, tetrahydrofolate. Abbreviations: MAT = methionine adenosyltransferase, MT = methyltransferase, SAAH = S-adenosylhomocysteine hydrolase, MS = methionine synthase, MTAFR = methylene tetrahydrofolate reductase.

**Figure 3 ijms-23-14908-f003:**
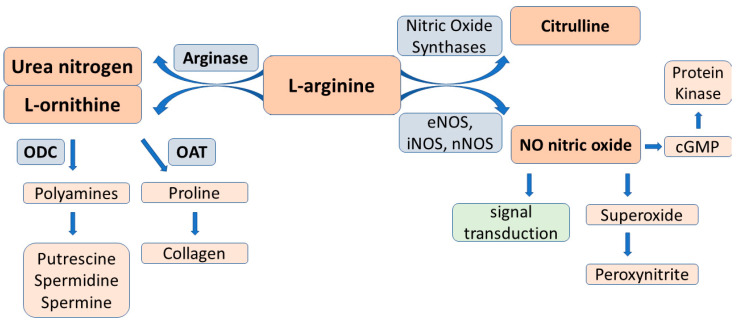
L-arginine catabolism pathways by nitric oxide synthases to nitric oxide and l-citrulline, and by arginase to l-ornithine and downstream metabolites. Abbreviations: eNOS, iNOS, nNOS → endothelial, inducible, and neuronal nitric oxide synthases; ODC = ornithine decarboxylase, OAT = ornithine aminotransferase.

**Figure 4 ijms-23-14908-f004:**
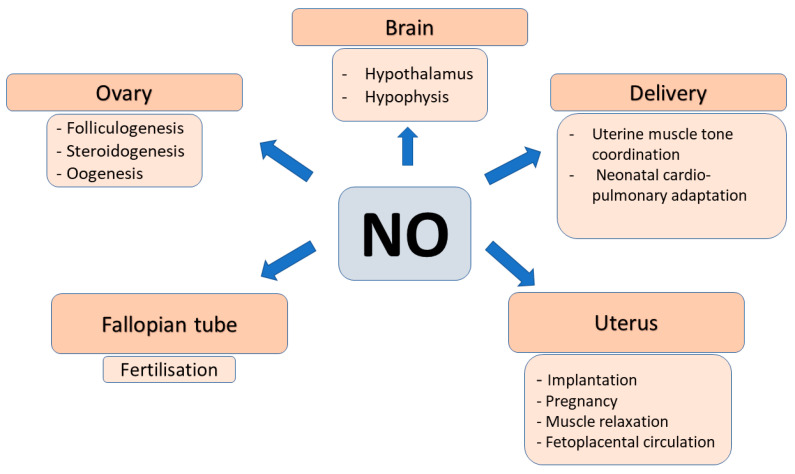
Summary figure for the involvement of nitrogen monoxide (NO) in the major processes of female reproduction.

## Data Availability

Not applicable.

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
