# Peer review of "The Role of L-Arginine-NO System in Female Reproduction: A Narrative Review"

_ijms, 2022, doi:10.3390/ijms232314908_

Round 1
Reviewer 1 Report
In the manuscript, “The Role of L-Arginine-NO System in Female Reproduction: A Narrative Review”, the authors aimed to review and summarize the current literature of the expression and function of the NO system during different stages of female reproduction. While there is merit in this work, I have a few comments that need to be addressed.
1. To me, Lines 37-43 in the introduction seemed out of place/too specific. It would be beneficial to move them to specific sections of the review or delete them entirely.
2. In section 2.4, it would be appropriate to add a brief discussion of nNOS and the brain’s role in these reproductive processes because it is certainly involved.
3. A summary figure of L-Arginine/NO and the reproductive processes discussed would be nice.
4. In Figure 1 on the far right, “Methyl” is spelled “Methil”.
5. There are a few grammatical/spelling errors throughout the document. These should be fixed.
Author Response
University of Pecs School of Medicine
Department of Obstetrics and Gynecology
Head: Kalman Kovacs, MD, Med. habil.
H-7624 Pecs, 17. Edesanyak u., Hungary
Tel: (0036)72 536 360;
Fax: (0036)72 536-000/#6360
Prof. Dr. Maurizio Battino,
Editor-in-Chief, International Journal of Molecular Sciences
Polytechnic University of Marche
Department of Specialised Clinical Sciences and Odontostomatology, Ancona, Italy
E-mail address: [email protected]
Tel.: +972-3-530-2882, +972-3-530-2784
Dear Assistant Editor Ms. Tamara Ugarković,
Please find our enclosed revised manuscript entitled, “The Role of L-Arginine-NO System in Female Reproduction: a Narrative Review”, which we are submitting to be considered for publication as a Review Article in the International Journal of molecular Sciences. All the reviewer’s comments have been considered, and the manuscript has been corrected accordingly.
Hereby I would like to provide a point-by-point response to each concern of the reviewers:
Reviewer number 1:
1.”To me, Lines 37-43 in the introduction seemed out of place/too specific. It would be beneficial to move them to specific sections of the review or delete them entirely”.
Thank you for your comment, according tot he suggestion we have deleted the requested part from the manuscript.
2.”In section 2.4, it would be appropriate to add a brief discussion of nNOS and the brain’s role in these reproductive processes because it is certainly involved.”
As it was requested the central role L-Arginine/NO system has been added as a new paragraph into the required section (2.4), with relevant references.
3.”A summary figure of L-Arginine/NO and the reproductive processes discussed would be nice”.
Following your suggestion, we created a summary figure, see figure 4 in the text.
4.”In Figure 1 on the far right, “Methyl” is spelled “Methil”.
Thank you for your observation, we have modified figure 1, into the right spelling.
- „There are a few grammatical/spelling errors throughout the document. These should be fixed”
We have carefully proofread the text and corrected the misspellings throughout the manuscript.
After edition based on the comments independent reviewers, we wish that this original article would be considered for publication in the International Journal of Molecular Sciences. The data have not been published elsewhere, nor is this manuscript under consideration at any other journal. If I can provide any other information, please contact me by phone, fax, e-mail, or at the following address.
We thank you in advance for your consideration of our manuscript and if any additional information is needed, please do not hesitate to contact me.
Sincerely,
Pecs, 23/11/2022. Dr. Balint Farkas, Med. habil.
Reviewer 2 Report
This review article summarized and focused on the L-arginine-No system relating to female reproduction. Both L-arginine and No are involved in several pathways in the cell. The molecular mechanism relationship to disease remains elusive. This review would help the readers to understand and reveal the mechanism and develop a therapy.
I found several issues described below.
(1) The authors drew two figures but did not refer to the main text. You have to cite them somewhere.
(2) It is helpful for the reader that the chemical structure of some compounds mentioned in the text, for example, MMA, ADMA, and SDMA, are drawn.
(3) Line 58
Na-+independent -> Na+-independent?
(4) Line 111
Please spell out ADMA and MMA
(5) Line 132
Generally, the term PRMT I and PRMT II are wrong descriptions. PRMTI and PRMT II should be type I PRMT and type II PRMT, respectively.
Author Response
University of Pecs School of Medicine
Department of Obstetrics and Gynecology
Head: Kalman Kovacs, MD, Med. habil.
H-7624 Pecs, 17. Edesanyak u., Hungary
Tel: (0036)72 536 360;
Fax: (0036)72 536-000/#6360
Prof. Dr. Maurizio Battino,
Editor-in-Chief, International Journal of Molecular Sciences
Polytechnic University of Marche
Department of Specialised Clinical Sciences and Odontostomatology, Ancona, Italy
E-mail address: [email protected]
Tel.: +972-3-530-2882, +972-3-530-2784
Dear Assistant Editor Ms. Tamara Ugarković,
Please find our enclosed revised manuscript entitled, “The Role of L-Arginine-NO System in Female Reproduction: a Narrative Review”, which we are submitting to be considered for publication as a Review Article in the International Journal of molecular Sciences. All the reviewer’s comments have been considered, and the manuscript has been corrected accordingly.
Hereby I would like to provide a point-by-point response to each concern of the reviewers:
Reviewer number 2:
1.”The authors drew two figures but did not refer to the main text. You have to cite them somewhere.”
The figures has been implemented into the text, and cited as requested.
- „It is helpful for the reader that the chemical structure of some compounds mentioned in the text, for example, MMA, ADMA, and SDMA, are drawn.”
A new figure demonstrating the chemical structure of the different methylargininase types has been added.
- „Line 58, Na-+independent -> Na+-independent?”
We corrected the text from Na-+ to Na+.
- „Line 111. Please spell out ADMA and MMA”
We spelled out each ADMA, MMA and SDMA in text.
- „Line 132 Generally, the term PRMT I and PRMT II are wrong descriptions. PRMTI and PRMT II should be type I PRMT and type II PRMT, respectively”
We have modified the text accordingly.
We have carefully proofread the text and corrected the misspellings throughout the manuscript.
After edition based on the comments independent reviewers, we wish that this original article would be considered for publication in the International Journal of Molecular Sciences. The data have not been published elsewhere, nor is this manuscript under consideration at any other journal. If I can provide any other information, please contact me by phone, fax, e-mail, or at the following address.
We thank you in advance for your consideration of our manuscript and if any additional information is needed, please do not hesitate to contact me.
Sincerely,
Pecs, 23/11/2022. Dr. Balint Farkas, Med. habil.